# A β-carotene-binding protein carrying a red pigment regulates body-color transition between green and black in locusts

**Meiling Yang[1†], Yanli Wang[2†], Qing Liu[1,3†], Zhikang Liu[1], Feng Jiang[4], Huimin Wang[4], Xiaojiao Guo[1], Jianzhen Zhang[2*], Le Kang[1,4*]**

[1]State Key Laboratory of Integrated Management of Pest Insects and Rodents, Institute of Zoology, Chinese Academy of Sciences, Beijing, China; [2]Institute of Applied Biology, Shanxi University, Taiyuan, China; [3]Sino-Danish College, University of Chinese Academy of Sciences, Beijing, China; [4]Beijing Institutes of Life Science, Chinese Academy of Sciences, Beijing, China

**Abstract** Changes of body color have important effects for animals in adapting to variable environments. The migratory locust exhibits body color polyphenism between solitary and gregarious individuals, with the former displaying a uniform green coloration and the latter having a prominent pattern of black dorsal and brown ventral surface. However, the molecular mechanism underlying the density-dependent body color changes of conspecific locusts remain largely unknown. Here, we found that upregulation of β-carotene-binding protein promotes the accumulation of red pigment, which added to the green color palette present in solitary locusts changes it from green to black, and that downregulation of this protein led to the reverse, changing the color of gregarious locusts from black to green. Our results provide insight that color changes of locusts are dependent on variation in the red β-carotene pigment binding to βCBP. This finding of animal coloration corresponds with trichromatic theory of color vision.

**\*For correspondence:**
zjz@sxu.edu.cn (JZ);
lkang@ioz.ac.cn (LK)

[†]These authors contributed equally to this work

**Competing interests:** The authors declare that no competing interests exist.

## Introduction

Body color change is a ubiquitous but highly diverse phenomenon of adaptive significance to animals under exposure to illnesses, environmental changes, predation, and sexual signals (*Burmeister et al., 2005*; *Henderson et al., 2017*; *Mäthger et al., 2003*; *Stuart-Fox and Moussalli, 2008*; *Teyssier et al., 2015*). Seasonal switching between color phases in response to extrinsic environmental factors is common in arthropods (*Futahashi and Fujiwara, 2008*; *Futahashi et al., 2012*; *Moran and Jarvik, 2010*). However, rapid and reversible color change is a rare trait regulated by endogenous and exogenous mechanisms (*Cuthill et al., 2017*; *Hubbard et al., 2010*). This type of phenotypic plasticity has not been well investigated.

Rapid body color changes in arthropods such as aphids, locusts and grasshoppers, lepidopteran caterpillars, and spider mites are influenced by environmental changes, especially by changes in population density (*Bryon et al., 2017*; *Tabadkani et al., 2013*; *Valverde and Schielzeth, 2015*; *Wang and Kang, 2014*; *Xiong et al., 2017*). Locust body color changes are of particular interest because conspecific color change coincides with changes in population density as well as behavioral changes; specifically, dramatic individual color change from green to black occurs when locusts form agriculturally destructive swarms (*Bolger et al., 2014*; *Wang and Kang, 2014*; *Wu et al., 2012*). Similar to that of other animals (*Cuthill et al., 2017*; *Duarte et al., 2017*), the coloration of locusts serves as a dynamic form of population information.

Phenotypic plasticity in the migratory locust *Locusta migratoria* underlies the transition between the gregarious and solitary phases, which are dependent on population density changes (*Guo et al., 2011*; *Jiang et al., 2017*; *Kang et al., 2004*; *Ma et al., 2011*; *Wu et al., 2012*; *Yang et al., 2014*). In addition to behavioral differences between the two phases of locusts (*Kang et al., 2004*; *Ma et al., 2011*; *Yang et al., 2014*), remarkable body color changes between gregarious and solitary locusts occur. Solitary locusts found at low population densities have a uniform green coloration, whereas gregarious locusts develop a striking color pattern of a black back with a contrasting brown ventral color (*Pener, 1991*; *Wang and Kang, 2014*).

The black-brown pattern of gregarious locusts is recognized as mainly being induced under high-density crowding conditions (*Pener, 1991*). Regarding the regulatory mechanism, juvenile hormones reportedly induce solitary green cuticular coloration in locusts (*Applebaum et al., 1997*; *Tanaka, 1993*). However, the regulatory function of these hormones in body coloration during phase change remains controversial (*Wang and Kang, 2014*). The neuropeptide [His7]-corazonin induces a dark coloration of the whole body in locusts but does not regulate the phase-related black–brown color pattern in gregarious locusts (*Tanaka, 2000*; *Wang and Kang, 2014*). Silencing the *ebony* and *pale* genes in the dopamine pathway that leads to the synthesis of dopamine melanin can result in partial fading of the black color in gregarious locusts, but the exogenous injection of dopamine does not completely lead to the black pattern (*Ma et al., 2011*). Thus, the body color pattern of gregarious locusts is not the result of pure melanization.

The green color of solitary locusts is likely due to the presence of a combination of yellow and blue pigments that facilitate the camouflage of low-density locusts in the background of plants (*Pener and Simpson, 2009*). In the present study, we investigated whether the black coloration of gregarious locusts results from the addition of pigments to the solitary green background.

We used strand-specific high-throughput RNA sequencing to monitor genome-wide transcriptional expression profiles and verify differentially expressed genes (DEGs) correlated with color polymorphism. We found a key DEG that encodes β-carotene-binding protein (βCBP), which promotes body color change associated with phase-related coloration. The manipulation of the gene expression by RNAi and feeding of β-carotene led to the reverse of body color of the locusts. The results of our study reveal that the black–brown coloration pattern in gregarious locusts is formed by the presence of the red color β-carotene pigment–βCBP protein complex in the green-colored background of solitary locusts. This work provides novel insights into the molecular mechanisms of body color changes in response to population density changes in accordance with the corresponding environmental changes.

## Results

### Gregarious and solitary locusts have different expression levels of βCBP

Gregarious and solitary locusts were reared either in crowded conditions or in solitude as previously described (*Ma et al., 2011*). These locusts displayed phase-specific phenotypes of body coloration. The gregarious locusts exhibited intense black patterns with a ventral brown color, and the solitary locusts showed the expected uniform green coloration (*Figure 1A*).

The body coloration of gregarious locusts is characterized as a black tergum of the thorax and abdomen, in particular, a very black pronotum. To delineate gene activity changes potentially associated with the regulation of phase-dependent body color traits, we performed transcriptome sequencing on pronotum integuments of gregarious and solitary locusts. We identified a total of 1653 DEGs between the gregarious and solitary locusts (*Figure 1—figure supplement 1A*), and 26% (430) of these DEGs were protein-coding genes (*Figure 1—figure supplement 1B*). Gene ontology analysis of these 430 protein-coding DEGs indicated that many of them encode cuticle metabolism-related proteins involved in chitin metabolic, melatonin metabolic, transport, and catecholamine catabolic activities (*Figure 1—figure supplement 1C*).

Sixty-eight of the 430 protein-coding DEGs are involved in pathways associated with animal coloration (*Figure 1B*). The top 17 DEGs (Log FC > 2.8, FDR < 1e-5) of these 68 genes were used to validate the differential expression patterns between the gregarious and solitary locusts via qPCR (*Figure 1C*). Nine of the 17 genes showed higher transcriptional levels in the pronotum integuments of the gregarious locusts than in those of the solitary locusts, and the transcriptional levels of 6 of

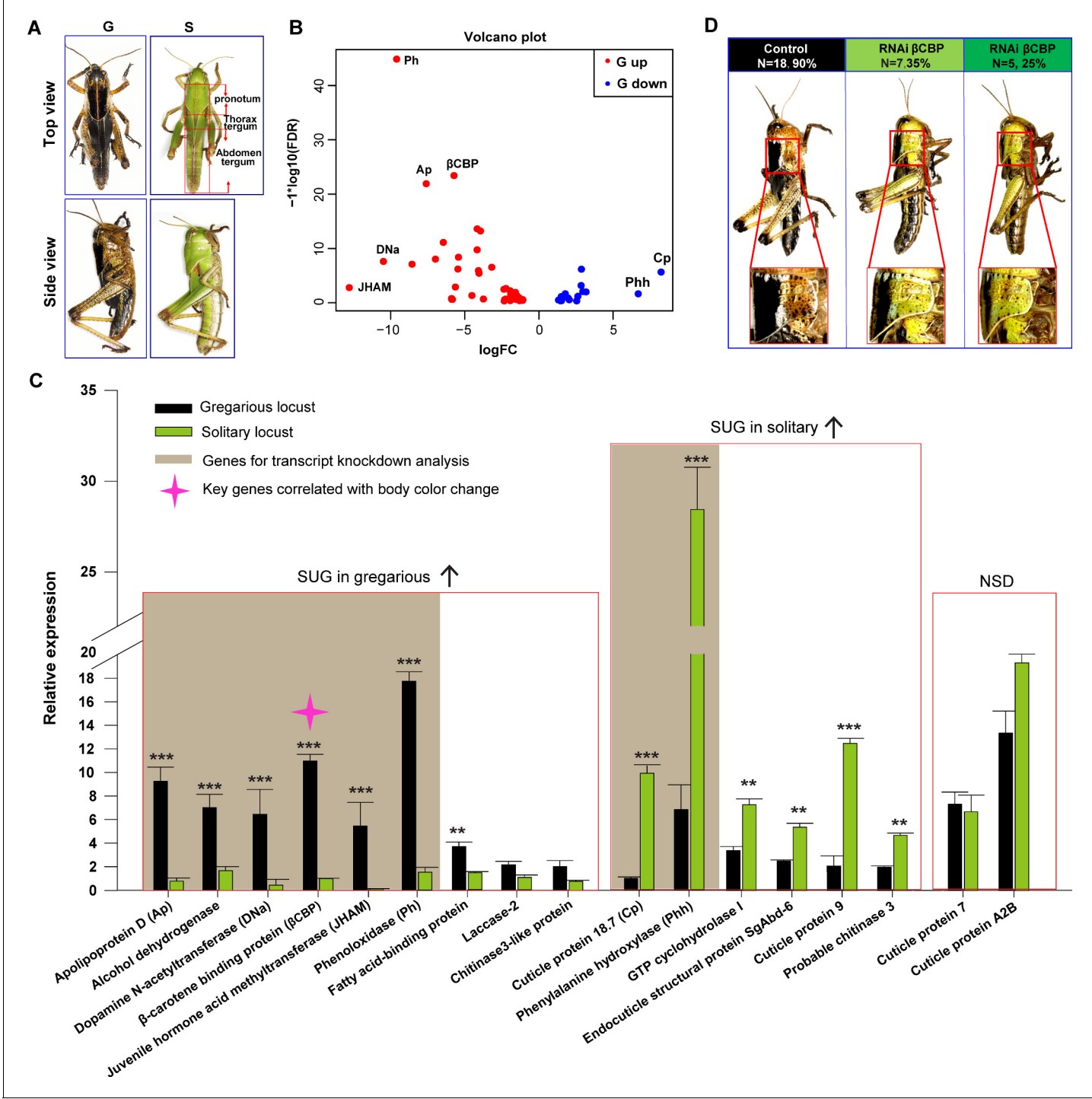

**Figure 1.** Identification of β-carotene-binding protein (βCBP) as a target associated with body color changes between gregarious and solitary locusts. (A) Body colors of typical gregarious and solitary nymphs. The tergums include pronotum, thorax tergum and abdomen tergum. (B) Logarithmic fold alterations of 68 differentially expressed genes (DEGs) associated with animal coloration between gregarious and solitary locusts are shown in the volcano plot diagram generated from genome-wide RNA-Seq. Red and blue dots indicate up- and down regulated genes, respectively, in the gregarious locusts (n = 2 samples, 10 integuments/sample). (C) The top 17 DEGs (Log FC >2.8, FDR < 1e−5) among the 68 genes from the volcano plot diagram were confirmed in the gregarious and solitary integuments via qPCR (n = 6 samples; eight integuments/sample; Student's t-test; red boxes indicate significantly upregulated genes (SUG) in gregarious/solitary locusts and no significant difference between gregarious and solitary locusts; *p < 0.05; **p < 0.01; ***p < 0.001). Brown shades denote the eight genes used for transcript knockdown analysis. The red star in the histogram indicate the identified target gene that correlates with body color change in the phase transition of locusts. (D) βCBP fosters the body color change

*Figure 1 continued on next page*

*Figure 1 continued*

from the gregarious pattern to the solitary pattern. Effects of βCBP dsRNA treatment on the body color of gregarious locusts were studied two stadiums after injection. Each second-instar nymph was injected with 3 μg of dsRNA three times on day three for the second-instar nymphs and on days 1 and 5 for the third-instar nymphs. Control nymphs were injected with equivalent volumes of dsGFP alone. .

The online version of this article includes the following source data and figure supplement(s) for figure 1:

**Source data 1.** Numerical data that are represented as graphs in *Figure 1C*.
**Figure supplement 1.** Transcriptomic profiles of locust pronotum integuments revealed by strand-specific RNA-seq.
**Figure supplement 2.** Analysis of interference efficiency of the other seven genes associated with animal coloration.

the 17 genes were significantly lower in the gregarious locusts than in the solitary locusts (*Figure 1C*). These results are consistent with the transcriptome data. Among the 17 genes, we investigated the top 8 DEGs (the difference fold of gene expression between gregarious and solitary locusts > 3), among which six had higher expression in the gregarious locusts and two had higher expression in the solitary locusts, to screen the key candidate genes involved in the color change.

To explore whether these eight genes regulate body color plasticity during locust phase transition, we carried out transcript knockdown analyses of these genes in the gregarious and solitary locusts. Out of the eight genes, only *βCBP* showed a significant effect on the regulation of body color transition although the other seven genes show a very significant decreased expression when they were silenced respectively (*Figure 1—figure supplement 2*). After molting, the dark pattern in the pronotum decreased markedly in 60% of the gregarious locusts after *βCBP* knockdown (*Figure 1D*). These data imply that *CBP* serves as a key gene involved in body color change.

## Phase-related characteristics of the *βCBP* gene in integument

To investigate the correlation of βCBP expression level with the degree of black body coloration of the locust integument, we performed quantitative reverse transcriptase–polymerase chain reaction (qRT–PCR) and found that *βCBP* was predominantly expressed in the locust integument tissue (*Figure 2—figure supplement 1*). Furthermore, consistent with the deeper black color of the pronotum than of the tergum of the thorax and abdomen in gregarious locusts, we found that *βCBP* expression was significantly higher in the pronotum than in the tergum integuments of the thorax and abdomen in gregarious locusts, whereas little *βCBP* expression was observed in the terga of solitary locusts, with no variability among the tergum segments (*Figure 2A*). Because the black coloration of gregarious locusts gradually increases with nymph developmental stage, we also quantified the *βCBP* transcript levels in the pronotum of different developmental stages of the gregarious locusts. The *βCBP* transcript level increased with developmental stage, with the expression levels of $4^{th}$ instar gregarious locusts being approximately 200 times those of the $2^{nd}$ and $3^{rd}$ instar gregarious locusts. By contrast, *βCBP* expression was invariable across the developmental stages in solitary locusts (*Figure 2B*). Therefore, the extent of black body coloration in locusts is positively correlated with *βCBP* expression level.

To explore whether *βCBP* is especially expressed in the black tergum integuments of gregarious locusts, we performed a comparative analysis via qRT–PCR to quantify the *βCBP* transcript levels in the tergum integuments of solitary and gregarious locusts. The transcript level of *βCBP* was significantly higher in the pronotum and tergum integuments of the thorax and abdomen of gregarious individuals than the pronotum and tergum integuments of solitary individuals (*Figure 2C* and *Figure 2—figure supplement 3*). Moreover, isolating the gregarious locusts (IG) significantly decreased the transcript level of *βCBP* (*Figure 2C* and *Figure 2—figure supplements 2* and *3*). By contrast, crowding the solitary locusts (CS) significantly increased the transcript level of *βCBP* (*Figure 2C* and *Figure 2—figure supplement 3*).

Because the expression pattern of *βCBP* in the pronotum was consistent with that in the tergum of the thorax and abdomen, the pronota were considered representative of all terga of the locusts in the following analyses. We also found that the level of CBP protein in the locust pronotum integuments was significantly higher in the gregarious locusts than in the solitary locusts (*Figure 2D*). Moreover, the protein levels of βCBP declined when the locusts were isolated and increased under the crowding conditions (*Figure 2D*). This expression pattern suggests a potential role of βCBP in

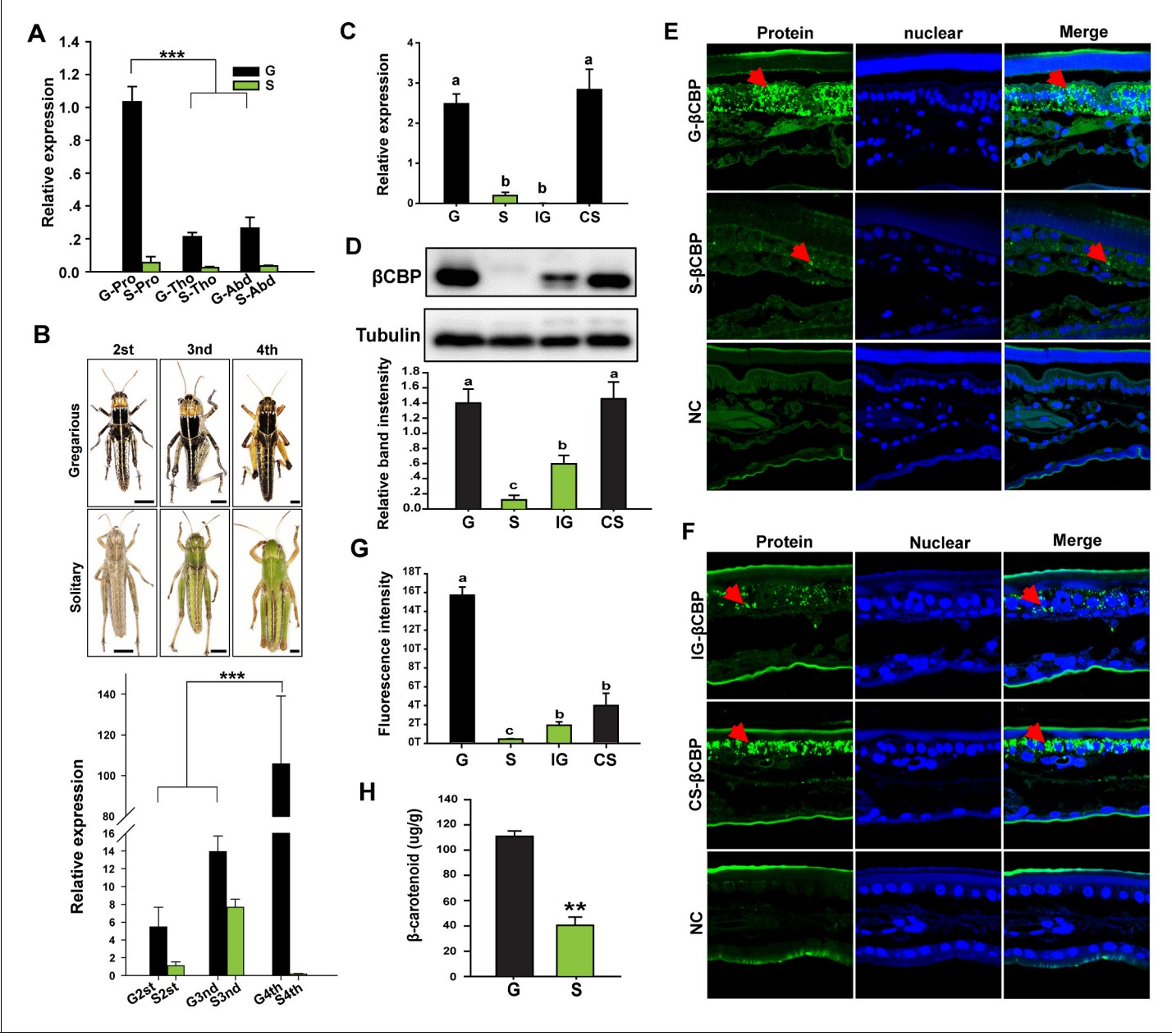

**Figure 2.** Expression or distribution of βCBP and its binding β-carotene shows phase-specific patterns. (A) *βCBP* expression in the tergums of different body segments of locusts in the gregarious (G) and solitary (S) phases. Pro: Pronotum; Tho: Thorax; Abd: Abdomen; (Tukey's test, ***p < 0.001). (B) *βCBP* expression in the pronotums of different instars of gregarious (G) and solitary (S) nymphs. Body color phenotypes of typical gregarious and solitary nymphs in the second (2st), third (3nd), and fourth (4th) instar are shown. The scale is 2 mm for each nymph. mRNA expression of *βCBP* was quantified by using qRT–PCR. qPCR data are shown as the means ± SEM (n = 4, ***p < 0.001). (C, D) mRNA (C) and protein (D) expression levels of βCBP were determined in the integuments of fourth-instar gregarious (G) nymphs, solitary (S) nymphs, gregarious nymphs after isolation (IG), and solitary nymphs after crowding (CS) using qPCR and Western blot analyses. Western blot bands were quantified using densitometry, and the values are expressed as the mean ± SEM (n = 3). (E) Distribution analysis of CBP was conducted to determine the localization and differences in abundance of CBP in the integuments between the gregarious (G) and solitary (S) locusts via immunohistochemistry. The red arrows indicate the areas where βCBP was localized in the locust integuments. NC, negative control. Images were visualized using an LSM 710 confocal fluorescence microscope (Zeiss) at a magnification of 40×. (F) Expression signals for βCBP were analyzed in the integuments of gregarious nymphs after isolation (IG) and solitary nymphs after crowding (CS). (G) Fluorescence intensity was quantified using ZEN 2.1 software and expressed as the means ± SEM (n = 3). (H) β-carotene content in the gregarious and solitary locust integuments was evaluated using HPLC (n = 6, **p < 0.01). Double comparisons were evaluated using Student's *t*-test and one-way ANOVA followed by Tukey's test was used for multiple comparisons. The same letter indicates that the data are not significantly different.

*Figure 2 continued on next page*

*Figure 2 continued*

The online version of this article includes the following source data and figure supplement(s) for figure 2:

**Source data 1.** Numerical data that are represented as graphs in *Figure 2A,B,C,D,G,H*.
**Figure supplement 1.** *βCBP* expression in the different tissues of gregarious locusts as determined by qRT–PCR.
**Figure supplement 2.** *βCBP* expression in the time course of gregarious locusts.
**Figure supplement 3.** *βCBP* expression in the tergum of the thorax and abdomen of locusts.

regulating the body color characteristics of the pronotum and tergum integuments of the thorax and abdomen.

To determine whether βCBP is differentially localized in the locust pronotum integuments of gregarious and solitary locusts, we performed immunohistochemistry of βCBP. We found that large amounts of βCBP were widely detected in the intercellular space of the pronotum of gregarious locusts, whereas βCBP was distributed very sporadically in the solitary locusts (*Figure 2E*). The amount of βCBP protein in the pronotum of gregarious locusts was 39.5-fold higher than the corresponding amount in solitary locusts (*Figure 2G*). Furthermore, the amount of βCBP protein in the locust pronotum was decreased by isolation and increased by crowding (*Figure 2F and G*).

Considering that βCBP can bind and accumulate β-carotenes in tissues (*Bhosale and Bernstein, 2007*), we assessed the amount of β-carotene binding by βCBP in the pronotum integuments of gregarious and solitary locusts in vivo. The β-carotene content in the gregarious locusts was 2.8-fold higher than that in the solitary locusts (*Figure 2H*). These results suggest that βCBP protein level and the associated β-carotene content are directly correlated with the degree of black coloration of the locust pronotum.

## βCBP mediates body color changes associated with population density

Based on the high level of βCBP available to bind with β-carotenes in gregarious locusts or crowded solitary locusts, we hypothesized that the black gregarious locusts could be induced toward green body color by blocking βCBP synthesis to interfere with β-carotene deposition. To confirm the involvement of βCBP in pigmentation during the phase transition, we injected double-stranded RNAs (dsRNAs) against the *βCBP* gene into the abdominal hemocoel of gregarious nymphs at the second stadium, isolated the injected locusts, and assessed their body color pattern.

After injection of *dsβCBP*–RNA and isolation treatment, the mRNA and protein levels of βCBP in the pronotum integument decreased by 99% and 85%, respectively, relative to the levels in *dsGFP*–RNA-injected and isolated controls (*Figure 3A and C*). In addition, the mRNA levels of *βCBP* in the tergum of the thorax and abdomen decreased by 98% relative to the levels in *dsGFP*–RNA-injected and isolated controls (*Figure 3—figure supplement 1*). The silencing effect of *dsβCBP*–RNA resulted in an 80% decrease in the signal intensity of βCBP in the pronotum integuments of gregarious locusts relative to the corresponding intensity in the *dsGFP*–RNA injected controls (*Figure 3E and F*).

We then determined the effects of *βCBP* silencing on the β-carotene content in the pronotum integuments of the locusts. Because there are many free β-carotenes in the locusts and β-carotene only binds with the βCBP protein to participate in the formation of body color (*Reszczynska et al., 2015*), we measured the amount of β-carotene binding to βCBP. The administration of *dsβCBP* significantly reduced the β-carotene content in the pronotum integuments from 104.8 µg/g to 74.3 µg/g (*p < 0.05, Figure 3H*).

A body coloration assay revealed that after molting of 29 locusts, 48% of the nymphs injected with *dsβCBP*–RNA and then isolated showed substantial lightening of the black pronotum. Moreover, 41% of the gregarious nymphs shifted to a complete green-colored body. By contrast, 73% of the *dsGFP*–RNA-injected gregarious nymphs treated with isolation retained an intense black body pattern. Only 27% of the *dsGFP*-injected and isolated control locusts showed body color lightening (*Figure 4A*). These results support the view that βCBP regulates the body color changes of gregarious and solitary locusts between black and green coloration.

To further confirm the direct involvement of βCBP in mediating body color changes, we assessed the effect of increasing the pigment bound with βCBP on the body color change of the locusts from solitary to gregarious phase. In this experiment, solitary locusts were fed a diet containing β-

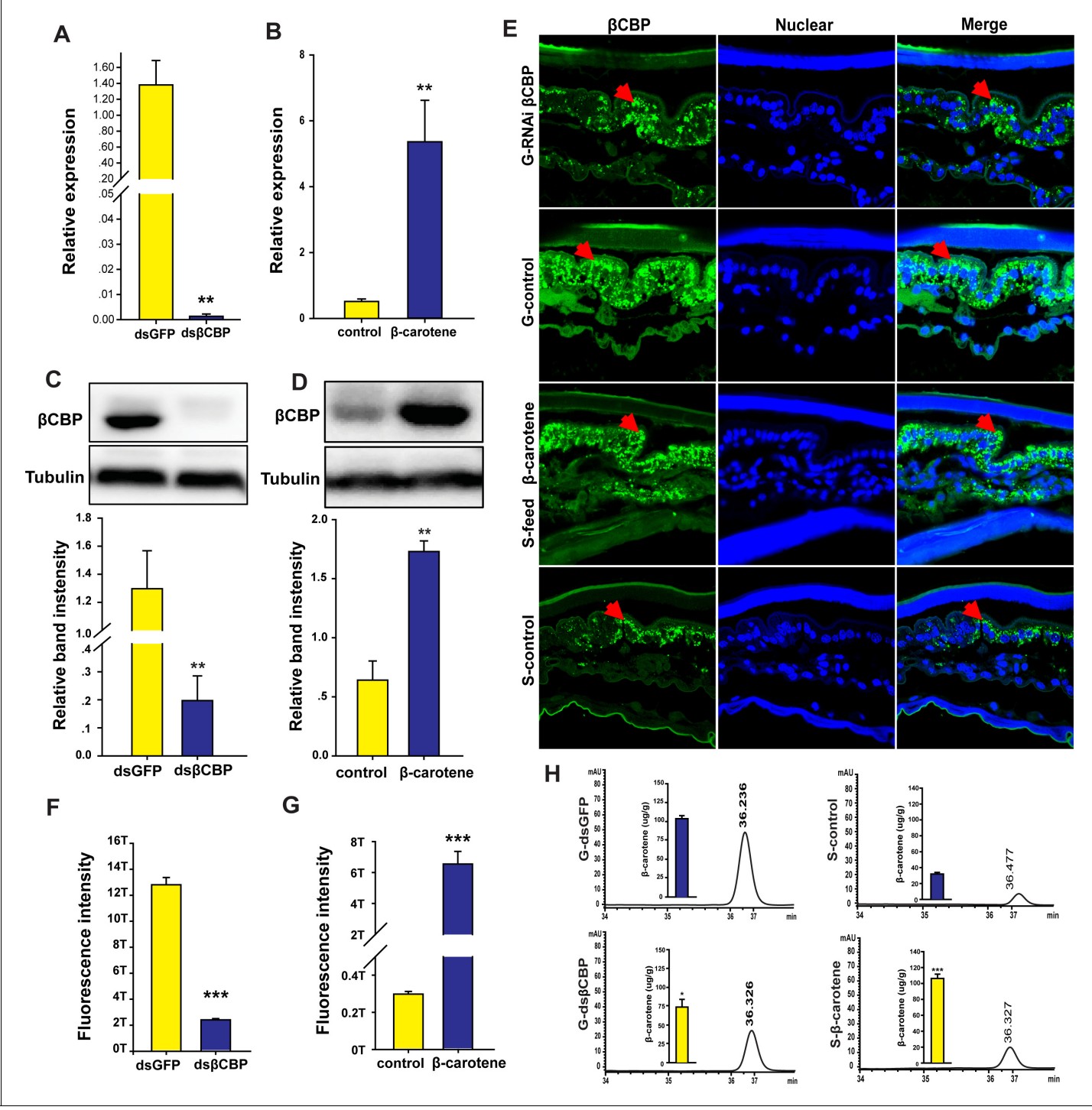

**Figure 3.** βCBP accumulates β-carotene pigment to affect pigmentation in the phase transition of locusts. (A, C) mRNA and protein expression of βCBP in the pronotum integument was quantified using qRT–PCR (A) and western blot analyses (C) two stadiums after βCBP silencing and isolating in gregarious locusts. (B, D) mRNA and protein expression levels of βCBP in the pronotum integument were determined two stadiums after feeding with β-carotene and crowding in solitary locusts using qPCR (B) and western blot analyses (D). (E) Fluorescence signals of βCBP were determined in gregarious (G) locusts after βCBP silencing coupled with isolation and in solitary (S) locusts after feeding with β-carotene diet coupled with crowding. (F, G) Fluorescence signal intensity of βCBP was quantified using ZEN 2.1 software and expressed as the means ± SEM (n = 3). (H) β-carotene content was evaluated in gregarious locusts after βCBP silencing coupled with isolation and in solitary locusts after feeding with β-carotene diet coupled with crowding by using HPLC. qPCR and HPLC data are shown as the means ± SEM (n = 6). Western blot bands were quantified using densitometry, and
*Figure 3 continued on next page*

*Figure 3 continued*

the values are expressed as the means ± SEM ($n$ = 3). All the double comparisons were evaluated using Student's $t$-test, *p < 0.05; **p < 0.01; ***p < 0.001.

The online version of this article includes the following source data and figure supplement(s) for figure 3:

**Source data 1.** Numerical data that are represented as graphs in *Figure 3A,B,C,D,F,G,H*.
**Figure supplement 1.** *βCBP* expression in the tergum of the thorax and abdomen of locusts after RNAi *βCBP*.
**Figure supplement 2.** βCBP expression in the tergum of the thorax and abdomen of solitary locusts.
**Figure supplement 3.** *βCBP* expression in the pronotum of solitary locusts only pre-fed with β-carotene but no crowding by qRT–PCR.

carotene and treated with crowding. The mRNA and protein levels of βCBP significantly increased by 11.2- and 2.5-fold in the pronotum, respectively, and increased by 2.3- and 2.1-fold in the tergum of the thorax and abdomen, respectively, relative to the corresponding levels in the regular diet-fed and crowded controls (*Figure 3B,D* and *Figure 3—figure supplement 2*). Moreover, the signal intensity of βCBP increased by 22.7-fold in the pronotum integument cells of the solitary locusts after β-carotene feeding and crowding relative to the intensity in the regular diet-fed and crowded controls (*Figure 3E and G*). In addition, HPLC analysis revealed that the amount of βCBP–β-carotene complex in the solitary integuments increased by 3.4-fold after β-carotene feeding and crowding (*Figure 3H*). A total of 47% of the solitary locusts completely shifted to the typical gregarious body color, and the pronota of 33% of the feeding locusts developed extensive black areas similar to the gregarious body color after feeding with β-carotene and crowding. However, 87% of the locusts retained their green color in the regular diet and crowding treatment (control group) (*Figure 4B*). Thus, βCBP and β-carotene can jointly induce body color changes from green to black associated with phase transition in the locusts.

To further determine whether the βCBP upregulation induced by β-carotene is responsible for the body color change of locusts from green to black, we conducted a rescue experiment by injecting *dsβCBP*-RNA into solitary locusts pre-fed with β-carotene. After molting, 35.5% of the *dsβCBP*-injected nymphs pre-fed with β-carotene showed green body color, whereas 15.6% of the green individuals in the *dsGFP*-injected group pre-fed with β-carotene did so. Correspondingly, the phenotype of black tergum individuals induced by β-carotene was rescued only by injection with *dsβCBP*-RNA (*Figure 4C*). Therefore, βCBP acts with β-carotene to regulate the black body color of gregarious locusts, and β-carotene by oneself did not regulate the body color.

## Body color change from green to black is mediated by the red pigment complex

To confirm the role of the βCBP–pigment complex in black color pigmentation, we performed a pigment immunoprecipitation assay using an antibody against the βCBP protein in vitro and in vivo to examine the binding capacity and complex coloration of βCBP with β-carotene (*Figure 5A and B*). We incubated recombinant βCBP (rβCBP) with β-carotene under immunoprecipitation by using βCBP antibody conjugated with protein A-Sepharose (*Figure 5A*). Immunoprecipitation of β-carotene-rβCBP formed a red complex, whereas no red precipitate formed when β-carotene was added to bovine serum albumin (BSA) or when β-carotene in the absence of rβCBP was incubated in the binding buffer followed by immunoprecipitation (*Figure 5A*). It could be inferred that the red precipitate was specific to the rβCBP–β-carotene complex because this complex failed to form in the presence of BSA and β-carotene or in the absence of rβCBP (*Figure 5A*). The pigment immunoprecipitation assay in vivo also showed an accumulation of red pigment in the precipitate of the integument when treated with anti-βCBP-immunoprecipitation compared with IgG-immunoprecipitation (*Figure 5B*). Furthermore, we confirmed that the red pigment in the βCBP precipitant result from β-carotene by HPLC analysis (*Figure 5—figure supplement 1*). These results provide strong evidence that βCBP directly binds with β-carotene in the pronotum integument to form the red complex, which directly contributes to the black pattern of the pronotum in gregarious locusts.

## βCBP is localized to the pigment granules of the pronotum integument

To verify the co-location of βCBP and the pigment granules, we investigated the subcellular distribution of βCBP using immunoelectron microscopy. The pronotum integuments were fixed and

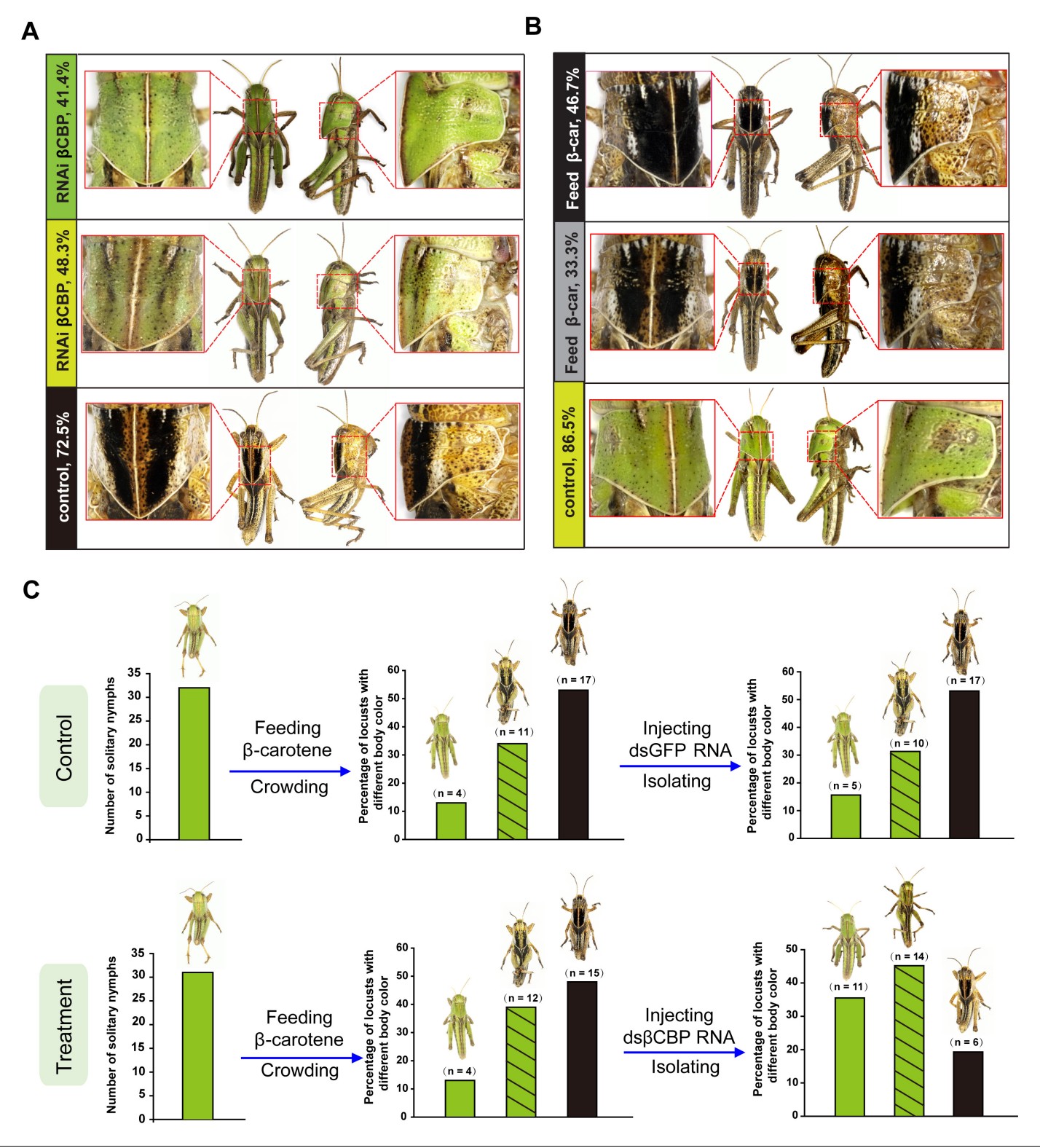

**Figure 4.** βCBP and its binding of β-carotene fosters the body color phase transition of the locust. (**A**) Effects of *βCBP* dsRNA treatment on the body color of gregarious locusts. Each second-instar nymph was injected with 3 µg of dsRNA thrice and exposed to isolation. Control nymphs were injected with equivalent volumes of dsGFP and then isolated. (**B**) Effects of β-carotene feeding on the body color of solitary locusts. Solitary nymphs were fed a synthetic diet containing β-carotene and were then subjected to crowding. Control insects were fed a regular diet without β-carotene and were then subjected to crowding. (**C**) A body color rescue experiment in solitary locusts was performed by injecting dsRNA against *βCBP* at day three in third-

*Figure 4 continued on next page*

*Figure 4 continued*

instar nymphs (N3D3) that had been pretreated with β-carotene feeding at day one as second-instar nymphs (N2D1). The control group was injected with dsGFP as third-instar nymphs that had been pretreated with β-carotene feeding as second-instar nymphs.

embedded for thin sections. The sections were analyzed for βCBP localization by using an anti-βCBP antibody followed by a gold-conjugated antibody against rabbit IgG. The gold particles were present mainly in the pigment granules of the pronotum integuments. In addition, the average number of gold particles in the pigment granules from the pronotum integuments was 15.5-fold higher in the gregarious locusts than in the solitary locusts (*Figure 5C and F*). Furthermore, *βCBP* silencing coupled with isolation caused a severe deficiency of gold particles in the pigment granules of the gregarious locusts, indicating a substantial reduction in the βCBP protein–pigment complex relative to the level in the *dsGFP*-injected and isolated controls (*Figure 5D and G*). By contrast, β-carotene intake coupled with crowding evidently increased the gold particles responsible for the βCBP production in the pigment granules of pronotum in the solitary locusts, demonstrating a significant promotion of βCBP protein–pigment complex accumulation compared with accumulation in the regular diet-fed and crowded controls (*Figure 5E and H*). Thus, the βCBP–pigment complex is a key mediator of black formation in the pronotum integuments of gregarious locusts.

## Discussion

Our study revealed an undescribed mechanism of body color regulation in which changes in the βCBP–β-carotene complex-mediated red color balance produce a shift between black and green body color in locusts. The change of body color occurs as an adaptive response to population density change. In high density, the gregarious locusts adopt the black-brown color pattern as alarm coloration against the predation by natural enemies, because they inevitably expose themselves to the environment. The green solitary locusts can hide themselves through the color confusion with the background of plants. The black–brown pattern in the gregarious locusts resulted from the superposition or balancing effect of a red pigment on the green background of solitary locusts rather than complete melanogenesis. Crowding treatment increased the βCBP levels in the integuments and caused body color change in the solitary locusts by the binding of βCBP with β-carotenes (*Figure 6*). This finding of animal coloration corresponds precisely with the physically trichromatic rule.

We demonstrated that βCBP expression is not only positively correlated with the black color of the pronotum in gregarious locusts but also phase-related in response to population density changes. The βCBP expression in solitary locusts was not induced by merely feeding with β-carotene without crowding treatment (*Figure 3—figure supplement 3*). Thus, the body color changes of solitary locusts that were induced to form gregarious-like body coloration must rely on crowding treatment and feeding the β-carotene, because only crowding treatment can promote the βCBP expression for βCBP–β-carotene complex production. Thus, population density of locusts acts as necessary and sufficient condition to drive the changes of body coloration. Generally, body color changes are driven by gene changes to adapt with a broad range of environments. For instance, the melanocortin-1 receptor gene regulates melanin production associated with black hair and skin in human and mouse pigmentation responsed to ultraviolet-light (*D'Orazio et al., 2006*). The major honest signaling of red carotenoids in Zebrafinch (*Mundy et al., 2016*) is thought to be the beak, colored in orange thanks to the ketolase activity encoded by the cytochrome P450 gene which was recruited from an ancestral function in retinal oil droplets. Although several studies have investigated the role of gene-environment in animal coloration, convergent phenotypes commonly arise from a single pigment-mediated independent change. Our study reveals a different mechanism in which βCBP, as a red signaling, has an integrative and palette effect on black and green body color change depending on the population density in locusts.

In our study, single βCBP or β-carotene cannot produce pigment deposition in the black tergum integuments of the locusts. βCBP only combine β-carotene to form red pigment complex for producing the black pattern of gregarious locusts. Previous studies in other species indicated that pigment-binding protein can uptake and bind pigment to affect coloration (*Sakudoh et al., 2011*; *Sakudoh et al., 2007*; *Wang et al., 2014a*). In the silkworm *Bombyx mori*, the yellow cocoon color is determined by the carotenoid-binding protein gene, which controls the uptake of carotenoids in

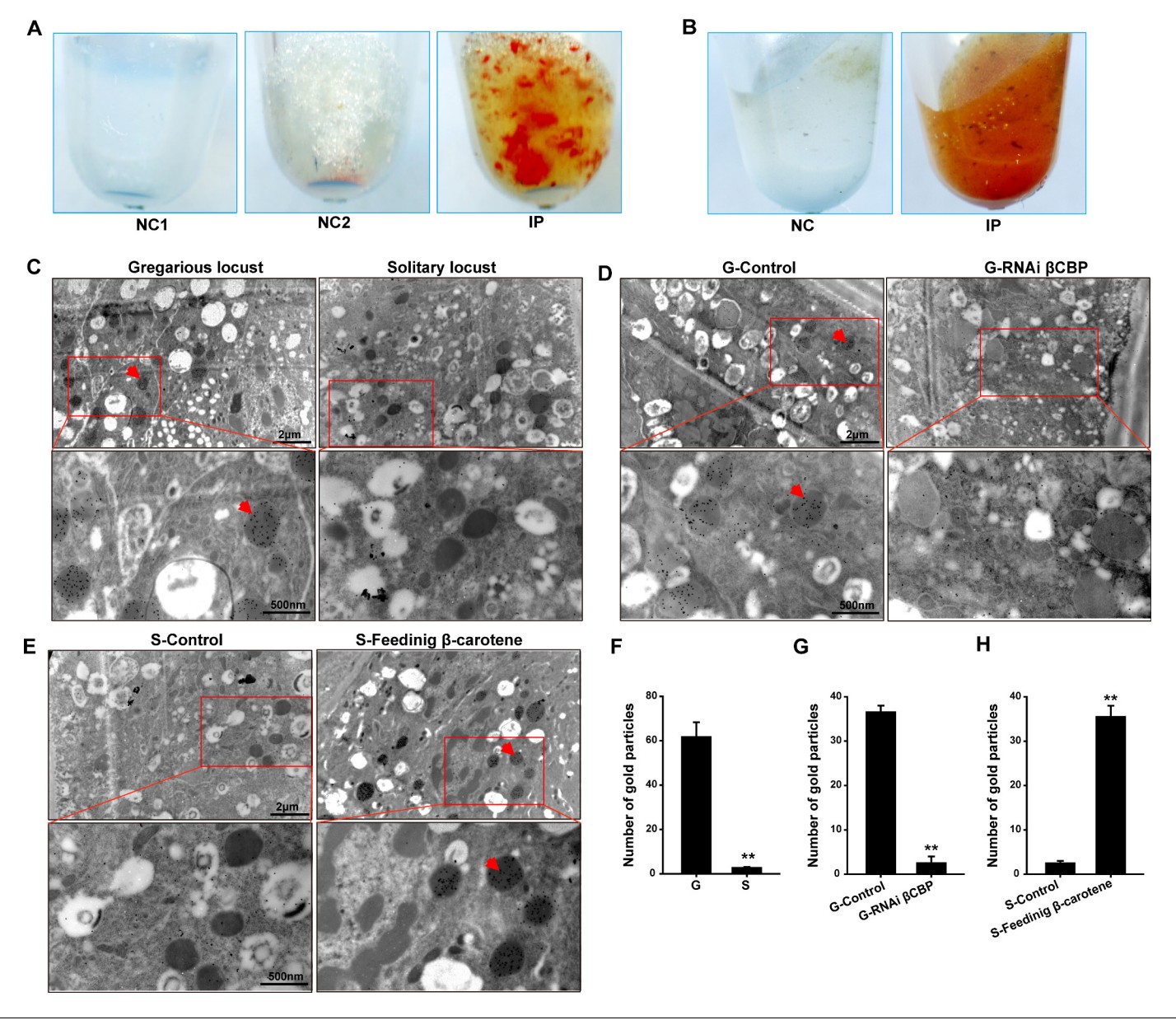

**Figure 5.** The red complex of βCBP located in the pigment granules of integument and βCBP binding of β-carotene directly contributes to the black back pattern of gregarious locusts. (A) Red βCBP–pigment complex was confirmed by recombinant βCBP (rβCBP) incubation with β-carotene under immunoprecipitation using βCBP antibody conjugated to protein A-Sepharose. NC1, containing β-carotene without rβCBP; NC2, containing BSA and β-carotene; IP, containing β-carotene and rβCBP. (B) Red pigment accumulates in the precipitate of the locust pronotum when treated with anti-βCBP-immunoprecipitation compared with IgG-immunoprecipitation. NC, containing pronotums and IgG; IP, containing pronotums and βCBP antibody. (C) The subcellular distribution of βCBP was investigated in the integuments of gregarious and solitary locusts by immunoelectron microscopy. (D, E) Immunogold labeling signals of βCBP were comparatively analyzed in the integuments of gregarious locusts after injection with *βCBP* dsRNAs and subsequent isolation and (D) in the integuments of solitary integuments after feeding with β-carotene diet followed by crowding (E). (F, G, H) Average number of gold particles in three randomly selected pigment granules in sections from various treatments. Variation is calculated based on three biological replicates (n = 3, Student's *t*-test, **p < 0.01). All sections were probed with anti-βCBP, followed by protein A-gold conjugate. Images outlined with red squares are magnified at 30,000 × using the Ruli H-750 TEM (Japan). The red arrow indicates the gold particles of βCBP located in the pigment granules.

The online version of this article includes the following source data and figure supplement(s) for figure 5:

**Source data 1.** Numerical data that are represented as graphs in *Figure 5F,G,H*.

**Figure supplement 1.** Confirmation of βCBP binding to β-carotene via HPLC.

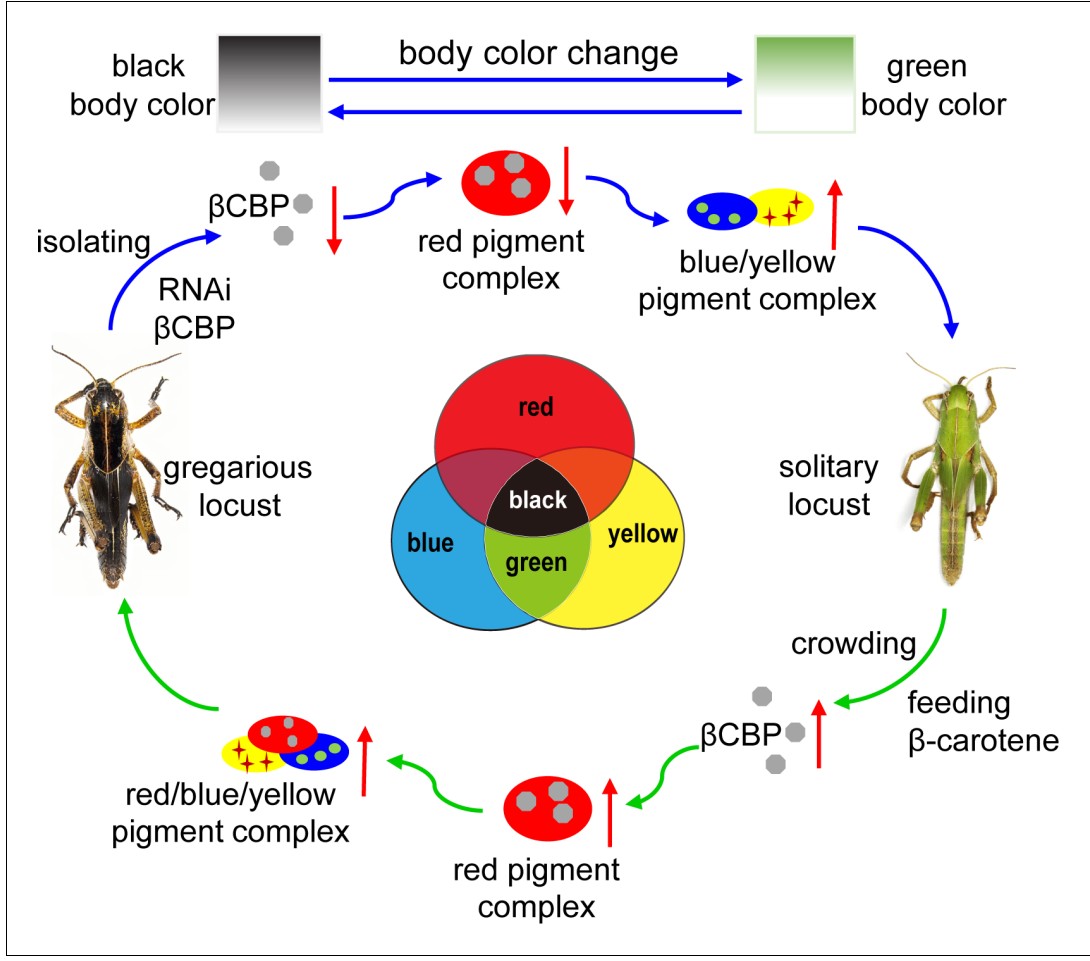

**Figure 6.** Model of a new mechanism in accordance with the physically trichromatic rule. Body color change is mediated by the red βCBP–β-carotene complex, which has a superposition effect on the solitary green background and produces the black back pattern of gregarious locusts. Silencing *βCBP* in gregarious locusts reduced the red color pigment β-carotene complex, thereby leading to the green pattern (instead of black). By contrast, feeding the solitary locusts β-carotene induced βCBP expression and caused gregarious-like body coloration.

The online version of this article includes the following figure supplement(s) for figure 6:

**Figure supplement 1.** *Biliverdin-binding protein* and *carotene-binding protein* expression in the pronotum of gregarious and solitary locusts by qRT–PCR.

the silk gland (*Sakudoh et al., 2007*). *Carotenoid-binding protein* mutation produces colorless hemolymph and white cocoons, indicating that carotenoid-binding protein plays a central role in cocoon color pigmentation (*Sakudoh et al., 2007*; *Wang et al., 2014a*). Mammals often obtain carotenoids by eating plants (*Avalos and Carmen Limón, 2015*; *Bryon et al., 2017*; *Toews et al., 2017*). Once absorbed, carotenoids are transported by lipoproteins, which are homologs of CBP in insects, through the bloodstream to the target tissues. In this process, the abundance and nature of lipoproteins limit the capacity to obtain carotenoids (*McGraw, 2006*). Interestingly, β-carotene-binding protein, as a key mediator of body color production, controls β-carotene content and absorption to produce red color, which is superposed on the background color of gregarious locus to yield the black pattern. In our present study, transcriptome analysis did not enrich the expression differentiation of catecholamine biosynthesis genes, which are considered involvement in the phase changes of the locust (*Ma et al., 2011*). We speculate the discrepancy mainly due to the different tissues sampling. In fact, they sampled the heads of the locusts (*Ma et al., 2011*), and our samples are from the pronotum of the locusts.

Our results show that the black pattern of gregarious locusts is caused by the interaction and mixture of different pigments. This pigment interaction has a colored palette effect that results in black coloration due to the accumulation of red color β-carotene pigments in the presence of blue and yellow pigments. The disappearance of the red pigment in the gregarious locusts led to the emergence of green color, which is the typical body color of solitary locusts. By elucidating the mechanism by which different types of pigments interact to create transitions among color polyphenism, this study provides new information that could be useful for analyzing color transitions in other organisms. In *Heliconius erato* and *Agraulis vanillae*, optix is required for the red color pattern and coordinates between the ommochrome (orange and red) and melanin (black and gray) pigment types (*Zhang et al., 2017*). Parrots acquire their colors through regulatory changes that drive the high expression of MuPKS in feather epithelia, which appear green when the blue and yellow pigments are present (*Cooke et al., 2017*). The green color of the integument in two species of locusts, the migratory locust and desert locust (*Schistocerca gregaria*) is obtained from a mixture of blue biliverdin pigment and yellow carotenoids (*Goodwin, 1952*; *Goodwin and Srisukh, 1951*). One of carotenoid binding proteins can bind with yellow pigments. For example, carotenoid binding protein plays a key role in the yellow cocoon pigmentation by binding with yellow carotenoids (*Tabunoki et al., 2004*). As expected, the expression levels of a *biliverdin-binding protein* and a *carotenoid-binding protein* in the pronotum integument were not significantly different between the gregarious and solitary locusts (*Figure 6—figure supplement 1*), suggesting that the contents of blue biliverdin and yellow carotenoid are the same in gregarious and solitary locusts. Thus, our results suggest a colored palette mechanism of body color change, with the change from typical gregarious-phased coloration (black) to solitary-phased coloration (green) is dependent on the extent of red β-carotene bound to βCBP.

The color polyphenism of gregarious and solitary locusts reflects an adaptive strategy to respond to changes in population density and the natural environment (*Simões et al., 2016*), which is consistent with the strategies of other animals when encountering different selection pressures (*Cuthill et al., 2017*). The multi-coloured pattern of black back with a contrasting brown ventral color in gregarious locusts functioned as effective warning coloration signal (*Sword et al., 2000*), providing mutual recognition to conspecifics and warning signal against predators in swarms (*Sword, 1999*). Potential predators, such as birds or and frogs (with tetrachromatic vision or trichromatic vision) could distinguish aposematic color visual signals as indicators of unpalatability and move on without an attempted attack. This phenomenon is consistent with other studies, showing that vertebrate predators more readily learn to avoid conspicuous prey (*Guilford and Cuthill, 1991*). Therefore, the expression of density-dependent warning coloration can result in a substantial decrease in predation on locusts in high-density population. This destabilization of predator-prey interactions could facilitate further population growth and gregarization which can ultimately lead to outbreaks. In addition, crowding stimuli, including the stress of a high population density and emission of an aggregative pheromone, may induce group migration (*Caro, 2009*; *Sword, 2002*). By contrast, the uniform cryptic color of solitary locusts may reduce the likelihood of detection and risk of predation, similar to the role of crypsis in other organisms (*Caro, 2009*; *Hemmi et al., 2006*). Animal coloration is a prominent phenotypical feature that serves multiple important functions, including social signaling, antipredator defense, parasitic exploitation, thermoregulation, and protection from ultraviolet light, microbes, and abrasion (*Bell et al., 2017*; *Caro et al., 2016*; *Cuthill et al., 2017*; *Duarte et al., 2017*; *Duarte et al., 2016*). Therefore, the mechanisms of color variation are highly conserved and can offer insights into the adaptive evolution of gene regulation. Our study shows a new 'palette effect' mechanism by which the red βCBP–β-carotene pigment complex can act as a switch to coordinate between black and green coloration. This mechanism of color pattern change may be evolutionarily conserved among multiple species, although additional functional work is needed to assess this hypothesis.

## Materials and methods

### Insects

All insects used in the experiments were reared in the same locust colonies at the Institute of Zoology, Chinese Academy of Sciences, Beijing, China. Gregarious nymphs were reared in large, well-

ventilated cages (400 insects per case). Solitary nymphs were cultured alone in white metal boxes supplied with charcoal-filtered compressed air. Both colonies were reared under a 14:10 light: dark photo regime at 30 ± 2°C (*Kang et al., 2004*; *Yang et al., 2016*). The newly hatched first instar nymphs are white but turns black in 1–2 hr. The head of second instar nymphs is larger and pale colour pattern is conspicuous. At the third instar nymph stage, two pairs of wing buds projects on each side of thorax. The color of fourth instar nymphs is conspicuously black.

## Strand-specific RNA sequencing of the integument

The pronotum integument was dissected from gregarious or solitary fourth-instar nymphs. Total RNA of the pronotum samples with two biological replicates for each group was extracted using TRIzol (Invitrogen) and treated with DNase I following the manufacturer's instructions. RNA quality was assessed using an Agilent 2100 Bioanalyzer (Agilent) to verify RNA integrity. cDNA libraries were prepared according to Illumina's protocols. The adaptor sequences in the raw sequencing data were filtered using Trimmomatic-0.30 (*Bolger et al., 2014*). We performed the de novo transcriptome assembly by using Trinity software (trinityrnaseq r20140717). Clean reads were mapped to the de novo transcriptome assembly in the accompanying script in Trinity software. The number of total reads was normalized using fragments per kilobase of transcripts per million reads mapped. The significance of differences between the test and control groups was based on *p* values with false discovery rate correction. DEGs with significance at corrected p < 0.05 in each comparison were determined using the run_DE_analysis in the R environment. The RNA-seq data were deposited in the Sequence Read Archive Database of NCBI (accession no. PRJNA399053).

## Assays of quantitative PCR for genes

Total RNA was isolated from the integument by using TRIzol (Invitrogen). The Moloney murine leukemia virus reverse transcriptase (Promega, USA) was used to prepare the oligo (dT)-primed cDNA. mRNAs were subjected to qPCR by using SYBR Green gene expression assays in accordance with the manufacturer's instructions (Tiangen). qPCR was performed on a LightCycler 480 instrument (Roche). *β-actin* was used as the internal control. Dissociation curves were determined for each mRNA to confirm unique amplification. The PCR efficiency of reference gene and *βCBP* were calculated (*Bustin et al., 2009*), which were within the commonly reported range of qRT-PCR. The mathematical model method was used to calculate the relative expression of genes (*Pfaffl, 2001*). The stability of three reference genes (*18S*, *GAPDH*, *β-actin*) were evaluated in different stage of gregarious and solitary locusts based on the gene-stability measure (*M*) values (*Supplementary files 1*) as calculated by geNorm algorithms (*Vandesompele et al., 2002*). The relative expressions were normalized using the geometric mean of two of most stable reference genes (*18S* and *β-actin*) with the low *M* values. The qPCR primers used are listed in *Supplementary files 2*. All the qRT–PCR reactions were performed in six biological replicates. Four integument samples were used in each biological replicate.

## βCBP immunohistochemical assay

βCBP was detected in the integument via immunohistochemistry. The integument was fixed in 4% paraformaldehyde overnight. Paraffin-embedded integument tissue slides (5 µm thick) were deparaffinized in xylene and rehydrated with an ethanol gradient. The samples were blocked with 5% BSA for 30 min and then incubated with an anti-βCBP antiserum (1: 200) for 2 hr. The slides were washed and incubated for 1 hr with Alexa Fluor-488 goat anti-rabbit secondary antibody (Life Technologies). Hoechst (1: 500) was used for nuclei staining. The βCBP signals were detected using an LSM 710 confocal fluorescence microscope (Zeiss).

## Transcript knockdown via RNAi

To knockdown the transcript of *βCBP*, dsRNA was synthesized using the T7 RiboMAXTM Express RNAi System (Promega, USA) following the manufacturer's instructions. Each insect was injected with 3 µg of dsRNA three times at day three for the second-instar nymphs and at days 1 and 5 for the third-instar nymphs. Control nymphs were injected with an equivalent amount of *dsGFP*–RNA. Nymphs that showed body color changes were used for the extraction. Six abnormal nymphs and six control nymphs were used for CBP immunohistochemistry.

### Feeding β-carotene assay

Solitary nymphs of 3-day-old second instars were fed with a synthetic diet containing β-carotene and subjected to crowding for two stadiums before calculating the body color phenotype statistics. The composition of the basic synthetic diet used in the experiment is shown in *Supplementary files 3*. Control insects were fed a synthetic diet without β-carotene and exposed to crowding. Rearing jars were cleaned and fresh diets were provided every day. A total of 29 nymphs were fed a synthetic diet in each of the treatment and control groups.

### Body color change experiment in vivo

To more quickly detect the changes of body coloration, we injected the double-strain RNA of $\beta CBP$ into the locusts and isolated the locusts. At same time we conducted the isolated locusts as the control. Thus, compared to the isolated control, we can obtain the true efficiency of RNA inferences. Because each injection of RNAi in general suppress effectively the target gene for several days, the isolation treatment can prevent the recovery expression of $\beta CBP$. In parallel, we fed with β-carotene for solitary locusts and crowded them. A body color rescue experiment was performed with solitary locusts. Approximately 30 second-instar nymphs were fed a synthetic diet containing β-carotene and then subjected to crowding. One stadium later, these nymphs were injected with 3 μg $ds\beta CBP$ or $dsGFP$ control and then isolated. One stadium after the dsRNA injection, the nymphs were subjected to body color analysis.

### Immunoelectron microscopy

For βCBP immunolabeling, integuments were fixed with 2.5% glutaraldehyde in 0.1 M phosphate buffer. After dehydration in a graded alcohol series, specimens were embedded in LR White resin and polymerized at 60˚C. Ultrathin sections were blocked with phosphate-buffered saline (PBS) containing 1% BSA, 0.2% Tween-20, and 0.2% gelatin before labeling with the CBP polyclonal antibody (1: 1000). A gold-conjugated (10 nm) secondary rabbit antibody was used to visualize the binding sites through TEM (Ruli H-750, Japan). The βCBP signals are detected by the black gold particles. Although there are some sporadic distribution in the background, the significant more gold particles were been observed in the pigment granules of locusts. The average number of gold particles was counted from three randomly selected pigment granules in sections of three biological replicates. The background signals were be counteracted in the treatment samples when compared with the controls.

### β-carotene binding assay in vitro and in vivo

A β-carotene-binding experiment in vitro was conducted as previously described (*Tabunoki et al., 2002*). Purified rβCBP was resuspended in PBS binding buffer containing 1 mg/mL BSA. Samples containing 50 M rCBP and 100 M β-carotene were incubated in 1 mL of binding buffer for 3 hr at 25˚C. The control experiments were performed using the same procedure without the use of purified rβCBP or BSA. Anti-CBP rabbit IgG–protein A-Sepharose was added to the reaction mixture and incubated for 4 hr at 4˚C. The protein A-Sepharose–rβCBP–β-carotene complex was collected by centrifugation at 13,000 g for 10 s. To remove the unbound β-carotene, the pellets were washed five times with 20 mM Tris-HCl (pH 7.6) containing 150 mM NaCl, 1% Nonidet P-40, protease inhibitor mixture, and 1 mg/mL BSA. Binding of β-carotene to rβCBP was confirmed by the presence of red-colored precipitate.

For the β-carotene binding experiment in vivo, approximately 200 pronotum integuments from gregarious individuals were collected and homogenized in ice-cold PBS buffer containing protease inhibitor. βCBP antibody or rabbit IgG, which was used as a negative control, and protein A-Sepharose was added to the homogenate and incubated at 4˚C overnight. The protein A-Sepharose-βCBP–β-carotene complex was collected by centrifugation at 13,000 g for 30 s. The pellets were washed five times to remove the unbound β-carotene.

### β-carotene quantification of the pronotum tissue extracts

The β-carotene content of locust pronotum integument was quantified using reverse-phase HPLC and DAD. Pronotum tissues of locust nymphs were rapidly dissected and stored in liquid nitrogen. Thirty pronotums were homogenized and lysed in lysate buffer. After incubation with βCBP

antibody–protein A-sepharose at 4℃ overnight, the specimens were placed in a 50-mL centrifuge tube containing a mixture of n-hexane, ethanol, and acetone (2: 1: 1, v: v: v). The tissue samples were sonicated at 5–10℃ for 15 min and then centrifuged at 6,800 g for 10 min. The upper layer extract and the ether extract of the lower layer residual solution were collected in another centrifuge tube. The same sample was re-extracted twice according to the same protocol as described above. Then, 2 mL of ether and 2 mL of distilled water were added to the lower layer collection tube and sonicated at 4℃ for 5 min. Following, the upper phase was collected. All of the extracts were combined and dried using a lyophilizer. The dried residue was dissolved in 2 mL of methyl tertbutyl ether (MTBE) and 2 mL of a KOH: methanol mixture (1: 9, w: v). After over 10 hr in darkness, 2 mL of MTBE and 2 mL of distilled water were added to the mixture. Then, the upper extract was collected and dried. This dried residue was dissolved in 200 µL of MTBE containing 0.1% BHT and filtered through 0.22 µm filters. The filtered samples were automatically loaded into the reverse phase HPLC system, which contained a carotene C30 column (250 mm $\times$4.6 mm, 5 µm, YMC, Japan), at a flow rate of 1 mL/min. The gradient elution method consisted of an initial 10 min of 71.2% acetonitrile, 23.8% methanol, and 5.0% H2O, followed by a linear gradient of 19.5% acetonitrile, 6.5% methanol, and 74.0% MTBE for 31 min. Data analysis was conducted with Agilent ChemStation software. β-carotene was quantified using an external standard.

## Statistical analysis

SPSS 17.0 software (SPSS, Inc.) was used for statistical analysis. Differences between treatments were evaluated using either Student's $t$-test or one-way ANOVA followed by Tukey's test for multiple comparisons. Body color transition (%) was compared between the treatment and control using Student's $t$-tests. For parametric analyses, the percentage data of body color transition were arcsin ($x$1/2) transformed, whereas the absolute quantity data were log ($x + 1$) transformed to correct heterogeneity of variances (*Wei et al., 2013*). A $p$ value of $< 0.05$ was considered statistically significant. All of the results are expressed as the means $\pm$ SEM.

## Acknowledgements

This work was funded by the Strategic Priority Research Program of the Chinese Academy of Sciences (CAS) (XDB11010200), National Key Plan for Scientific Research and Development of China (2016YFC1200603, 2017YFD0200406), Natural Science Foundation of China (NSFC 31430023, 31472051, 31672364, 31872302, 31802018), and Graduate Student Education Innovation Project of Shanxi Province (2016BY017).

## Additional information

### Funding

| Funder | Grant reference number | Author |
| --- | --- | --- |
| National Natural Science Foundation of China | 31430023 | Le Kang |
| National Natural Science Foundation of China | 31472051 | Meiling Yang |
| National Natural Science Foundation of China | 31672364 | Jianzhen Zhang |
| National Natural Science Foundation of China | 31872302 | Meiling Yang |
| National Natural Science Foundation of China | 31802018 | Yanli Wang |
| Graduate Student Education Innovation Project of Shanxi Province | 2016BY017 | Yanli Wang |
| Chinese Academy of Sciences | Strategic Priority Research Program - XDB11010200 | Le Kang |

| National Key Plan for Scientific Research and Development of China | 2016YFC1200603 | Meiling Yang |
| National Key Plan for Scientific Research and Development of China | 2017YFD0200406 | Meiling Yang |

The funders had no role in study design, data collection and interpretation, or the decision to submit the work for publication.

## Author contributions

Meiling Yang, Supervision, Project administration, Writing—review and editing; Yanli Wang, Data curation, Formal analysis, Validation, Writing—original draft, Writing—review and editing; Qing Liu, Zhikang Liu, Data curation, Formal analysis, Validation; Feng Jiang, Validation; Huimin Wang, Data curation; Xiaojiao Guo, Jianzhen Zhang, Resources; Le Kang, Supervision

## Author ORCIDs

Le Kang (iD) http://orcid.org/0000-0003-4262-2329

## Decision letter and Author response

Decision letter https://doi.org/10.7554/eLife.41362.sa1
Author response https://doi.org/10.7554/eLife.41362.sa2

# Additional files

## Supplementary files

• Supplementary file 1. The gene-stability measure ($M$) values of three reference genes in different stage of gregarious and solitary locustsas as calculated by geNorm algorithms.

• Supplementary file 2. Primers used in the qPCR analysis of $\beta CBP$, and *actin* and *GFP* used for RNA interference.

• Supplementary file 3. Composition of the basic synthetic diet.

• Transparent reporting form

## Data availability

The RNA-seq data were deposited in the Sequence Read Archive Database of NCBI (accession no. PRJNA399053).

The following dataset was generated:

| Author(s) | Year | Dataset title | Dataset URL | Database and Identifier |
|---|---|---|---|---|
| Meiling Yang, Yanli Wang, Qing Liu, Zhikang Liu, Feng Jiang, Huimin Wang, Xiaojiao Guo, Jianzhen Zhang, Le Kang | 2018 | A $\beta$-carotene-binding protein carrying a red pigment regulates body-color transition between green and black in locusts | http://www.ncbi.nlm.nih.gov/bioproject/399053 | NCBI Sequence Read Archive, PRJNA399053 |

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
