## [Decision Letter]

Thank you for submitting your article "Red and black: A β-carotene-binding protein carrying a red pigment regulates body-color transition in locusts" for consideration by *eLife*. Your article has been reviewed by three peer reviewers, including Shannon Olsson as the Reviewing Editor, and the evaluation has been overseen by K. VijayRaghavan as the Senior Editor. The following individuals involved in review of your submission have agreed to reveal their identity: Jean-Michel Gibert (Reviewer #2); Uwe Irion (Reviewer #3).

The reviewers have discussed the reviews with one another and the Reviewing Editor has drafted this decision to help you prepare a revised submission.

The proximate and ultimate mechanisms of coloration in the animal kingdom are of core interest to a large number of fields, from developmental biology to animal behavior. The striking coloration change of locusts while shifting to their gregarious phase has been a key attribute to identify their associated behavioral changes. As such, identifying the genetic and biochemical source of this coloration is of great interest and relevance.

Summary:

Here, the authors identify a specific β-carotene binding protein that becomes elevated in certain body parts during solitary-gregarious phase change. This binding protein, when complexed to β-carotene, produces a red pigment that on the standing green cuticular background becomes dark brown/black, giving gregarious locusts their unique patterning. The authors first performed a transcriptome analysis of the two morphs and identified 68 genes involved in animal coloration whose expression was modulated between morphs. They chose the top 17 of these genes for RT-qPCR validation and focus later on one of these genes: the gene encoding Β-carotene binding protein. The authors show convincingly using RNAi experiments, controlled diets, crowding/isolation conditions, immuno-staining and electron microscopy, that β-carotene and β-carotene binding protein act jointly for the deposition of β-carotene in the integument of the gregarious morphs. The authors find that expression of the gene encoding βCBP is significantly up-regulated in gregarious v. solitary locusts coinciding with a colour change to black/brown from green. Furthermore, they demonstrate that the β-carotene binding protein actually binds carotene and that knock-down of the gene leads to a (partial) body colour change back to green.

In general, the experiments are well performed, discussed, and appropriate for *eLife*. However, the reviewers have some important considerations to improve the clarity and accuracy of the results and conclusions.

Essential revisions:

1) Two-part titles and/or punctuation are to be avoided:

In the interests of style and clarity, the titles of *eLife* research papers should not use colons, dashes, exclamation marks, or brackets (unless needed for scientific reasons). Two-part titles should also be avoided (though some exceptions can be made for Tools and Resources). Please revise your title with this advice in mind.

2) RT-qPCR analyses:

First, only one gene (Β-actin) is used as an internal control. Standards require several genes to be used as internal controls. Furthermore, it is not explained how the relative expression levels are quantified. Precise methods using the calculated efficiency of the primers such as the Pfaffl method should be used. Kindly refer to Bustin et al., 2009 (The MIQEGuidelines: Minimum Information for Publication of Quantitative Real-Time PCR Experiments). The authors are advised to perform additional RT-qPCR on at least one more control gene and explain how expression levels were calculated for all RT-qPCR experiments.

Also, RT-qPCR primers are given in Supplementary file 1 for βCBP, and actin but not for the other 16 genes whose expression was quantified in Figure 1C. The primers used to measure the expression of biliverdin-binding protein gene and carotenoid-binding protein in Figure 6—figure supplement 1 are also missing. The sequences of missing primers should be provided.

3) Source of Pigmentation:

While the role of β-CBP is well supported, it may not be the only source of coloration. It would be good for the authors to acknowledge and clarify the role of other proteins. For example, The authors state that they carried out transcript knock-down analyses of the top 8 DEGs and found that only βCBP showed a significant effect. Please clarify what is meant by "Top 8", and clarify their criteria for inclusion of DEGs. The data for the knock-down efficiency should be provided for all transcripts; e.g. Phenoloxidase shows a very significant differential expression, but the knock-down has no effect on colouration. Importantly, the authors also state that they "…show that the black pattern of gregarious locusts is not caused by an increase in melanin…". If they want to uphold this statement they should determine whether melanin is present in the gregarious locusts, or if the black colour is indeed produced solely by the combination of yellow, blue and red pigments.

4) Previous studies:

There seems to be a discrepancy between the data in this current manuscript and those published by Ma et al., 2011, where catecholamine biosynthesis genes were found to be up-regulated in gregarious locusts. Here, it appears that in Figure 1—figure supplement 1 up-regulation of catecholamine catabolic, catechol-containing catabolic and dopamine catabolic GOs are reported.

5) Statistics:

Please provide more details for the statistics, including assumptions on normality and homoscedasticity, p-values, F, SS, SSmean and degrees of freedom for all cases. Please also indicate in figures which test was used (e.g. Figure 2 column H used Tukey post hoc tests). Furthermore, no statistical tests are performed to analyze some of the experiments, while terms such as "significant effect" are provided in the text (e.g. Figure 1—figure supplement 2; Figure 4A, B and C). Please clarify this with appropriate tests or correct wording.

6) Isolation experiments:

Please clarify the coupled injection of RNAi against BetaCBP and isolation of the injected individuals in experiments corresponding to Figure 4A and 4C. As both of these treatments should separately decrease the expression of βCBP gene, it is unclear why they are coupled.

7) Trichromatic Rule:

The trichromatic rule support is interesting, particularly when insects are not trichromatic. The authors should discuss this in relation to the potential role of this pigmentation to locust fitness. The authors note: "The contrasting body coloration of gregarious locusts may allow mutual recognition and warning against predators in swarms.". Since insects are not trichromatic, it would be good to acknowledge this and also discuss how these patterns might be seen by various animals (and why).

[Editors' note: further revisions were requested prior to acceptance, as described below.]

Thank you for resubmitting your work entitled "A β-carotene-binding protein carrying a red pigment regulates body-color transition between green and black in locusts" for further consideration at *eLife*. Your revised article has been favorably evaluated by K VijayRaghavan as the Senior Editor, a Reviewing Editor. And 2 reviewers.

The manuscript has been significantly improved and all reviewers agree that it provides a significant contribution to the field. However, there remain some remaining issues that need to be addressed before acceptance, as outlined below:

1) For RT-qPCR, the authors tested RP49 and GAPDH in addition to Β-actin for normalization. They write that Β-actin is the most stable gene because its M value is 0.488 and chose to use only this gene for normalization. However, the authors should report the M value all tested genes, and use the geometric mean of the expression of the two most stable control genes (or of the three control genes if these M values are not too distant from one another) as advised in Vandesompele et al., 2002. Furthermore, to calculate the relative expression it is recommended to use the Pfaffl method (Pfaffl, 2001) (cited in Bustin et al., 2009) as the method used in the manuscript (2-ΔΔCT) is known not to be reliable.

2) While the authors did address several of the criticisms presented in the initial review, these are clarified only in the rebuttal letter and not the text. All data and calculations should be also mentioned in the manuscript text (Results and/or Materials and methods) as well, e.g.: a) Add M values for all tested genes and calculations of expression (Rebuttal Letter Point 2).

b) Add explanation of discrepancy with Ma et al., 2011 (Rebuttal Letter Point 4).

c) Add explanation of coupled injection of RNAi against ΒCBP and isolation of injected individuals (Rebuttal Letter Point 6).

d) Please clarify what "warning" (i.e. which locust defense systems) the "warning coloration" provides a signal for with references (Rebuttal Letter Point 7).

3) We appreciate the authors attention to English, but there remain several small typos in the text, e.g.:

a) "…the other seven genes show a very significant…"

b) "significantly reduced" where is the statistical test here? Otherwise, remove the term significant.

c) – "…locusts adopt the black-brown color pattern as alarm coloration against the predation by natural enemies, because they inevitably expose themselves to the environment…"

d) "third instar nymph stage".

---

## [Author Response]

Essential revisions:1) Two-part titles and/or punctuation are to be avoided:In the interests of style and clarity, the titles of eLife research papers should not use colons, dashes, exclamation marks, or brackets (unless needed for scientific reasons). Two-part titles should also be avoided (though some exceptions can be made for Tools and Resources). Please revise your title with this advice in mind.

Thanks for this suggestion. We revised the manuscript title as “A β-carotene-binding protein carrying a red pigment regulates body-color transition between green and black in locusts”.

2) RT-qPCR analyses:First, only one gene (Β-actin) is used as an internal control. Standards require several genes to be used as internal controls. Furthermore, it is not explained how the relative expression levels are quantified. Precise methods using the calculated efficiency of the primers such as the Pfaffl method should be used. Kindly refer to Bustin et al., 2009 (The MIQEGuidelines: Minimum Information for Publication of Quantitative Real-Time PCR Experiments). The authors are advised to perform additional RT-qPCR on at least one more control gene and explain how expression levels were calculated for all RT-qPCR experiments.

We accepted the suggestion. We assessed three reference genes (*β-actin, RP49, GAPDH*) for their suitability in the locusts. Specificities of primers were confirmed by the presence of a single peak in melting curve analysis (see Author response image 1). The PCR efficiency of these candidate reference genes and βCBP ranged from 97%-103%, and the correlation coefficient (r^2^) ranged from 0.991 to 0.998, which were within the commonly reported range of qRT-PCR, according to the method described by Bustin et al. (Bustin et al., 2009). We compared the expression levels of these genes in different nymph stage of gregarious and solitary locusts. The expression stability of these genes was evaluated using geNorm algorithms (Vandesompele et al., 2002). All candidate genes were ranked based on the average expression stability values (M) values. By comparison, as shown in Author response image 2, gene expression stability, the most stable genes were *β-actin* (M = 0.488). So *β-actin* was used as the most suitable references gene for accurate normalization.

Amplification efficiency of primers

Gene namesPrimer sequenceAmplification efficiency (%)Correlation coefficient (r^2^)GAPDHF:GGCAGTTAATGACCCGTTCA R:ACAACAAGACTGTCGCCATC97.60.99218SF:ATGCAAACAGAGTCCCGACCAGA R:GCGCAGAACCTACCATCGACAG103.60.996β-actinF:AATTACCATTGGTAACGAGCGATT R: TGCTTCCATACCCAGGAATGA101.020.998βCBPF:GGTGGAGAAGTGGCTGGCTCAG R:GGGAAGACCGCCCTGTAGAAGC99.640.991

**Author response image 2. respfig2:** Gene expression stability of reference genes in different stage of gregarious and solitary locusts as calculated by geNorm (Vandesompele et al., 2002).

Also, RT-qPCR primers are given in Supplementary file 1 for βCBP, and actin but not for the other 16 genes whose expression was quantified in Figure 1C. The primers used to measure the expression of biliverdin-binding protein gene and carotenoid-binding protein in Figure 6—figure supplement 1 are also missing. The sequences of missing primers should be provided.

Thank you for your reminder. We added RT-qPCR primers of 16 genes in Figure 1C and biliverdin-binding protein gene and carotenoid-binding protein gene in Figure 6—figure supplement 1(see Supplementary file 1).

3) Source of Pigmentation:While the role of β-CBP is well supported, it may not be the only source of coloration. It would be good for the authors to acknowledge and clarify the role of other proteins. For example, The authors state that they carried out transcript knock-down analyses of the top 8 DEGs and found that only βCBP showed a significant effect. Please clarify what is meant by "Top 8", and clarify their criteria for inclusion of DEGs. The data for the knock-down efficiency should be provided for all transcripts; e.g. Phenoloxidase shows a very significant differential expression, but the knock-down has no effect on colouration. Importantly, the authors also state that they "…show that the black pattern of gregarious locusts is not caused by an increase in melanin…". If they want to uphold this statement they should determine whether melanin is present in the gregarious locusts, or if the black colour is indeed produced solely by the combination of yellow, blue and red pigments.

The "Top 8" means the difference fold of gene expression between gregarious and solitary locusts >3. We added it to the third paragraph of the subsection “Gregarious and solitary locusts have different expression levels of βCBP”.

We provided the interference efficiency of other 7 genes (Figure 1—figure supplement 2). Also, the statement "…show that the black pattern of gregarious locusts is not caused by an increase in melanin…" was revised as “Our results show that the black pattern of gregarious locusts is caused by the interaction and mixture of different pigments”.

4) Previous studies:There seems to be a discrepancy between the data in this current manuscript and those published by Ma et al., 2011, where catecholamine biosynthesis genes were found to be up-regulated in gregarious locusts. Here, it appears that in Figure 1—figure supplement 1 up-regulation of catecholamine catabolic, catechol-containing catabolic and dopamine catabolic GOs are reported.

Thanks for this comment. The *genes*are*expressed*ina*tissue-specific pattern*. In the paper by Ma et al., 2011, the head tissues of gregarious and solitary locusts were used for microarray analysis due to the *tissue*-*specific* high level of key genes involved in catecholamine metabolic pathway. They found that the genes pale and henna were expressed at a higher level in gregarious head tissues than in solitary head tissues by qRT-PCR. Here, we used the integuments tissue to screen the key candidate genes involved in the color change because *βCBP* was predominantly expressed in the locust integument tissue. The catecholamine catabolic genes showed low and no different expression in integuments tissue (*pale: p*=0.18427, fdr=1; *henna: p*=0.45347, fdr=1). Consequently, this data are not discrepancy compared to the result in the paper by Ma et al., 2011. On the other hand, our previous study (Ma et al., 2011) indicated that the silences of the genes in dopamine pathway can make the black pallescent, do not result in body color change from black to green.

5) Statistics:Please provide more details for the statistics, including assumptions on normality and homoscedasticity, p-values, F, SS, SSmean and degrees of freedom for all cases. Please also indicate in figures which test was used (e.g. Figure 2 column H used Tukey post hoc tests). Furthermore, no statistical tests are performed to analyze some of the experiments, while terms such as "significant effect" are provided in the text (e.g. Figure 1—figure supplement 2; Figure 4A, B and C). Please clarify this with appropriate tests or correct wording.

We provided the more details for the statistics, including assumptions on normality and homoscedasticity, p-values, F, SS, in the supporting data (see supplementary Figure 1—source data 1, Figure 2—source data 2, Figure 3—source data 1 and Figure 5—source data 1).

We also added the statistics test in the Figure 2, 3, and 4 legend. Furthermore, the statistical test of body color transition were provided in the “Statistical analysis” section as: “Body color transition (%) was compared between the treatment and control using Student’s *t*-tests. For parametric analyses, the percentage data of body color transition were arcsin (*x*1/2) transformed, whereas the absolute quantity data were log (x + 1) transformed to correct heterogeneity of variances.”

6) Isolation experiments:Please clarify the coupled injection of RNAi against βCBP and isolation of the injected individuals in experiments corresponding to Figure 4A and 4C. As both of these treatments should separately decrease the expression of βCBP gene, it is unclear why they are coupled.

Thank you for the suggestion. Population density of locusts acts as necessary and sufficient condition to drive the changes of body coloration. The black pattern in the pronotum of gregarious locusts decreased in 60% of the gregarious locusts only after *βCBP* knockdown without isolation treatment (Figure 1D). However, the silencing effect of *dsβCBP*–RNA coupled with isolation resulted in an 80% decrease in the signal intensity of βCBP in the pronotum integuments of gregarious locusts relative to the corresponding intensity in the *dsGFP*–RNA injected and isolated controls (Figure 4A). If not isolation, crowding will offset the low expression level of CBP after RNAi in gregarious locusts. Thus, the dramatic body color changes of gregarious locusts that were induced to form solitary-like body coloration in short time must rely on isolation treatment and RNAi CBP manipulation, because isolation treatment can always maintain the low expression level of *CBP*.

7) Trichromatic Rule:The trichromatic rule support is interesting, particularly when insects are not trichromatic. The authors should discuss this in relation to the potential role of this pigmentation to locust fitness. The authors note: "The contrasting body coloration of gregarious locusts may allow mutual recognition and warning against predators in swarms.". Since insects are not trichromatic, it would be good to acknowledge this and also discuss how these patterns might be seen by various animals (and why).

We added some discussion as the following:

“In high density, the gregarious locusts adopt the black-brown color pattern as alarming coloration against the predation of natural enemies, because they inevitably expose to the environment. The green solitary locusts can hide themselves through the color confusion with the background of plants.”

“Potential predators, such as birds or and frogs (with tetrachromatic vision or trichromatic vision) could distinguish aposematic color visual signals as indicators of unpalatability and move on without an attempted attack. […] This destabilization of predator-prey interactions could facilitate further population growth and gregarization which can ultimately lead to outbreaks.”

[Editors' note: further revisions were requested prior to acceptance, as described below.]

The manuscript has been significantly improved and all reviewers agree that it provides a significant contribution to the field. However, there remain some remaining issues that need to be addressed before acceptance, as outlined below:1) For RT-qPCR, the authors tested RP49 and GAPDH in addition to Β-actin for normalization. They write that Β-actin is the most stable gene because its M value is 0.488 and chose to use only this gene for normalization. However, the authors should report the M value all tested genes, and use the geometric mean of the expression of the two most stable control genes (or of the three control genes if these M values are not too distant from one another) as advised in Vandesompele et al., 2002. Furthermore, to calculate the relative expression it is recommended to use the Pfaffl method (Pfaffl, 2001) (cited in Bustin et al., 2009) as the method used in the manuscript (2-ΔΔCT) is known not to be reliable.

Thanks for this suggestion. We added the *M* value of all tested reference genes in the Supplementary file 1. The relative expressions of genes were calculated by the mathematical model method (Pfaffl, 2001) and normalized using the geometric mean of two of most stable reference genes (*18S* and *β-actin*) with the low *M* values as described previously (Vandesompele et al., 2002). (See the Supplementary file 1, Figure 1C, 2A, 2B, 2C, 3A, 3B; Figure 1—figure supplement 2, Figure 2—figure supplement 1, Figure 2—figure supplement 2, Figure 2—figure supplement 3, Figure 3—figure supplement 1, Figure 3—figure supplement 2A,Figure 3—figure supplement 3, Figure 6—figure supplement 1). The mathematical model method and utilization of the reference genes are cited in the Materials and methods as the following:

“The PCR efficiency of reference gene and *βCBP* were calculated (Bustin et al., 2009), which were within the commonly reported range of qRT-PCR. […] The relative expressions were normalized using the geometric mean of two of most stable reference genes (*18S* and *β-actin*) with the low *M* values.”

2) While the authors did address several of the criticisms presented in the initial review, these are clarified only in the rebuttal letter and not the text. All data and calculations should be also mentioned in the manuscript text (Results and/or Materials and methods) as well, e.g.: a) Add M values for all tested genes and calculations of expression (Rebuttal Letter Point 2).

We added the *M* values for all tested reference genes and calculated the relative expression according to the mathematical model method suggested by reviewers (See the Supplementary file 1, Figure 1C, 2A, 2B, 2C, 3A, 3B; Figure 1—figure supplement 2, Figure 2—figure supplement 1, Figure 2—figure supplement 2, Figure 2—figure supplement 3, Figure 3—figure supplement 1, Figure 3—figure supplement 2A,Figure 3—figure supplement 3, Figure 6—figure supplement 1). The mathematical model method and utilization of the reference genes are cited in the Materials and methods (subsection “Assays of quantitative PCR for genes”).

b) Add explanation of discrepancy with Ma et al., 2011 (Rebuttal Letter Point 4).

We added the explanation of discrepancy in Discussion as the following:

“In our present study, transcriptome analysis did not enrich the expression differentiation of catecholamine biosynthesis genes, which are considered involvement in the phase changes of the locust (Ma et al., 2011). We speculate the discrepancy mainly due to the different tissues sampling. In fact, they sampled the heads of the locusts (Ma et al., 2011), and our samples are from the pronotum of the locusts.”

c) Add explanation of coupled injection of RNAi against Β CBP and isolation of injected individuals (Rebuttal Letter Point 6).

We added the explanation of coupled injection of RNAi against β-CBP and isolation of injected individuals as the following:

“To more quickly detect the changes of body coloration, we injected the double-strain RNA of *βCBP* into the locusts and isolated the locusts. […] Because each injection of RNAi in general suppress effectively the target gene for several days, the isolation treatment can prevent the recovery expression of *βCBP*. In parallel, we fed with β-carotene for solitary locusts and crowded them.”

d) Please clarify what "warning" (i.e. which locust defense systems) the "warning coloration" provides a signal for with references (Rebuttal Letter Point 7).

We added the explanation as the following:

“The multi-coloured pattern of black back with a contrasting brown ventral color in gregarious locusts functioned as effective warning coloration signal (Sword et al., 2000), providing mutual recognition to conspecifics and warning signal against predators in swarms (Sword, 1999).”

3) We appreciate the authors attention to English, but there remain several small typos in the text, e.g.:a) "…the other seven genes show a very significant…"

Revised as required (subsection “Gregarious and solitary locusts have different expression levels of βCBP”).

b) "significantly reduced" where is the statistical test here? Otherwise, remove the term significant.

Revised as required (subsection “βCBP mediates body color changes associated with population density”).

c) "…locusts adopt the black-brown color pattern as alarm coloration against the predation by natural enemies, because they inevitably expose themselves to the environment…"

Revised as required (Discussion, first paragraph).

d) "third instar nymph stage".

Revised as required (subsection “Insects”).